# SELECTING THE BEST IN GANS FAMILY: A POST SELECTION INFERENCE FRAMEWORK

**Yao-Hung Hubert Tsai**[*1] , **Denny Wu**[* 2], **Ruslan Salakhutdinov**[1]
{[1]Machine Learning Department, [2]Computational Biology Department} @ Carnegie Mellon University
yaohungt@cs.cmu.edu, yiwu1@andrew.cmu.edu ,rsalakhu@cs.cmu.edu

**Makoto Yamada**[* 3,4], **Ichiro Takeuchi**[4,5],**Kenji Fukumizu**[3,6]
[3]RIKEN AIP, [4]JST PRESTO, [5]Nagoya Institute of Technology, [6]Institute of Statistical Mathematics
makoto.yamada@riken.jp, takeuchi.ichiro@nitech.ac.jp, fukumizu@ism.ac.jp

## ABSTRACT

"Which Generative Adversarial Networks (GANs) generates the most plausible images?" has been a frequently asked question among researchers. To address this problem, we first propose an *incomplete* U-statistics estimate of maximum mean discrepancy $\text{MMD}_{inc}$ to measure the distribution discrepancy between generated and real images. $\text{MMD}_{inc}$ enjoys the advantages of asymptotic normality, computation efficiency, and model agnosticity. We then propose a GANs analysis framework to select and test the "best" member in GANs family using the Post Selection Inference (PSI) with $\text{MMD}_{inc}$. In the experiments, we adopt the proposed framework on 7 GANs variants and compare their $\text{MMD}_{inc}$ scores.

## 1 INTRODUCTION

Despite the success of Generative Adversarial Networks (GANs) for generating plausible samples, the qualitative evaluation of the model performance remains a crucial issue. Numerous approaches have been proposed; however, most of them failed to provide meaningful scores Huang et al. (2018). For example, Inception Scores (Salimans et al., 2016) and Mode Scores (Che et al., 2016) measure the quality and diversity of the generated samples, but they were not able to detect overfitting and mode dropping/ collapsing for generated samples. The Frechet Inception Distance (FID) (Heusel et al., 2017) defines a score using the first two moments of the real and generated distributions, whereas the Classifier Two-Sample Tests (Lopez-Paz & Oquab, 2016) considers the classification accuracy of a binary classifier as a statistic for two-sample testing. Although the two metrics perform well in terms of discriminability, robustness, and efficiency, they require the distances between samples to be computed in a suitable feature space. We can also use Kernel density estimation (KDE) to estimate the density of a distribution; or more recently, Wu et al. (2016) proposed to apply annealed importance sampling (AIS) to estimate the likelihood for the decoder-based generative models. Nevertheless, these approaches need the access to the generative model for computing the likelihood, which are less favorable comparing to the model agnostic approaches which rely only on a finite generated sample set. Maximum Mean Discrepancy (MMD) (Gretton et al., 2012), on the other hand, has less weakness and is preferred against its competitors (Sutherland et al., 2016; Huang et al., 2018).

To measure the distribution discrepancy between generated and real images, in this paper, we introduce an incomplete U-statistics estimator $\text{MMD}_{inc}$, which has a number of compelling properties: asymptotic normality, computation efficiency, and model agnosticity. Then, we propose a hypothesis testing framework based on the Post Selection Inference (PSI) and $\text{MMD}_{inc}$ for GANs analysis. The framework is able to find a member in GANs family with the most plausible generated samples and test whether the selected member is able to generate samples that cannot be differentiated from the real distribution.

## 2 PROPOSED GANS ANALYSIS FRAMEWORK

Suppose we are given independent and identically distributed (i.i.d.) samples $\boldsymbol{X}^{(s)} = \{\boldsymbol{x}_i^{(s)}\}_{i=1}^n \in \mathbb{R}^{d \times n}$ from a $d$-dimensional distribution $p_s$ and $s \in \{1, \ldots, S\}$. Similarly, we have i.i.d. samples

---
[*]Equal contribution. Random author ordering.

$\boldsymbol{Y} = \{\boldsymbol{y}_j\}_{j=1}^n \in \mathbb{R}^{d \times n}$ from another $d$-dimensional distribution $q$. In particular, for GANs analysis, $\boldsymbol{x}_i^{(s)}$ is a feature vector generated by $s$-th GAN model with random seed $i$ and $\boldsymbol{y}_j \in \mathbb{R}$ is a feature vector of an original image. Image features can be pixel values or be extracted by pre-trained neural networks such as Resnet (He et al., 2016). Our goal is to first find a GAN model that generates samples closest to the real distribution and then test if $p_k = q$, where $k$ is the index of the selected GAN model.

## 2.1 Incomplete U-statistics MMD Estimator as GANs Evaluation Metric

The complete U-Statistics estimator of MMD (Gretton et al., 2012) is defined as

$$\mathrm{MMD}_u^2[\mathcal{F}, \boldsymbol{X}, \boldsymbol{Y}] = \frac{1}{n(n-1)} \sum_{i \neq j} h(\boldsymbol{u}_i, \boldsymbol{u}_j),$$

where

$$h(\boldsymbol{u}, \boldsymbol{u}') = k(\boldsymbol{x}, \boldsymbol{x}') + k(\boldsymbol{y}, \boldsymbol{y}') - k(\boldsymbol{x}, \boldsymbol{y}') - k(\boldsymbol{x}', \boldsymbol{y})$$

is the U-statistics kernel for MMD, $k(\boldsymbol{x}, \boldsymbol{x}')$ is a kernel function, and $\boldsymbol{u} = [\boldsymbol{x}^\top \ \boldsymbol{y}^\top]^\top \in \mathbb{R}^{2d}$. Although $\mathrm{MMD}_u$ has been sample efficient and model agnostic for GANs evaluation (Sutherland et al., 2016; Huang et al., 2018), it suffers from the computation inefficiency ($O(n^2)$ complexity), and its degenerated Null distribution creates a challenge for hypothesis testing.

To address the issues, we propose to use an *incomplete* U-statistics MMD (Wu et al., 2017) estimator:

$$\mathrm{MMD}_{inc}^2[\mathcal{F}, \boldsymbol{X}, \boldsymbol{Y}] = \frac{1}{\ell} \sum_{(i,j) \in \mathcal{D}} h(\boldsymbol{u}_i, \boldsymbol{u}_j),$$

where $\mathcal{D}$ is an arbitrary subset of $\{(i,j)\}_{i \neq j}$ and $\ell$ is $|\mathcal{D}|$. Under the condition that $\lim_{n,\ell \to \infty} n^{-2}\ell = 0$, $\mathrm{MMD}_{inc}$ is asymptotically normal (can be proved using Corollary 1 of Janson (1984)). Empirically, we choose $\ell = r \cdot n$ where $r$ is a small integer, and thus the computation complexity of $\mathrm{MMD}_{inc}$ is $O(n)$ which is computationally efficient. In particular, the specific design of $\mathcal{D} = \{(1,2), (3,4), \ldots, (n-1, n)\}$ corresponds to the linear-time MMD estimator (Gretton et al., 2012).

To sum up, as an alternative to $\mathrm{MMD}_u$, $\mathrm{MMD}_{inc}$ enjoys the benefit of Normal asymptotic distribution, computation efficiency, sample efficiency, and model agnosticity. Note that in addition to GANs evaluation metric, $\mathrm{MMD}_{inc}$ can also be adopted in MMD-based works such as MMD GAN (Li et al., 2017) and ReViSE (Tsai et al., 2017).

## 2.2 GAN analysis with MMDINF

Next, we propose to use `mmdInf` (Wu et al., 2017) as a hypothesis testing framework for selecting the "best" GAN that generates the samples closest to the real distribution. By integrating $\mathrm{MMD}_{inc}$, we formulate the hypothesis test as follows:

- $H_0$: $\mathrm{MMD}_{inc}^2[\mathcal{F}, \boldsymbol{X}^{(k)}, \boldsymbol{Y}] = 0 \mid k$-th GAN generates samples closest to the real distribution,
- $H_1$: $\mathrm{MMD}_{inc}^2[\mathcal{F}, \boldsymbol{X}^{(k)}, \boldsymbol{Y}] \neq 0 \mid k$-th GAN generates samples closest to the real distribution.

We employ the Post Selection Inference (PSI) framework to test the hypothesis.

**Theorem 1** *(Lee et al., 2016) Suppose that $\boldsymbol{z} \sim \mathcal{N}(\boldsymbol{\mu}, \boldsymbol{\Sigma})$, and the feature selection event can be expressed as $\boldsymbol{Az} \leq \boldsymbol{b}$ for some matrix $\boldsymbol{A}$ and vector $\boldsymbol{b}$, then for any given feature represented by $\boldsymbol{\eta} \in \mathbb{R}^n$ we have*

$$F_{\boldsymbol{\eta}^\top \boldsymbol{\mu}, \boldsymbol{\eta}^\top \boldsymbol{\Sigma} \boldsymbol{\mu}}^{[V^-(\boldsymbol{A}, \boldsymbol{b}), V^+(\boldsymbol{A}, \boldsymbol{b})]}(\boldsymbol{\eta}^\top \boldsymbol{z}) \quad | \quad \boldsymbol{Az} \leq \boldsymbol{b} \sim \mathrm{Unif}(0, 1),$$

*where $F_{\mu, \sigma^2}^{[a,b]}(x)$ is the cumulative distribution function (CDF) of a truncated normal distribution truncated at [a,b], and $\Phi$ is the CDF of standard normal distribution with mean $\mu$ and variance $\sigma^2$. Given that $\boldsymbol{\alpha} = \boldsymbol{A}\frac{\boldsymbol{\Sigma}\boldsymbol{\eta}}{\boldsymbol{\eta}^\top \boldsymbol{\Sigma}\boldsymbol{\eta}}$, the lower and upper truncation points can be computed by*

$$V^-(\boldsymbol{A}, \boldsymbol{b}) = \max_{j: \boldsymbol{\alpha}_j < 0} \frac{\boldsymbol{b}_j - (\boldsymbol{Az})_j}{\boldsymbol{\alpha}_j} + \boldsymbol{\eta}^\top \boldsymbol{z}, \ \ V^+(\boldsymbol{A}, \boldsymbol{b}) = \min_{j: \boldsymbol{\alpha}_j > 0} \frac{\boldsymbol{b}_j - (\boldsymbol{Az})_j}{\boldsymbol{\alpha}_j} + \boldsymbol{\eta}^\top \boldsymbol{z}.$$

**Marginal Screening with Discrepancy Measure:** Assume we have an estimate of MMD for each GAN: $\boldsymbol{z} = [\text{MMD}_{inc}^2[\mathcal{F}, \boldsymbol{X}^{(1)}, \boldsymbol{Y}], \dots, \text{MMD}_{inc}^2[\mathcal{F}, \boldsymbol{X}^{(S)}, \boldsymbol{Y}^{(S)}]]^\top \in \mathbb{R}^S \sim \mathcal{N}(\boldsymbol{\mu}, \boldsymbol{\Sigma})$. We denote the selected index by $k$ and the index set of the unselected GANs $\bar{\mathcal{S}}$. Since we want to test the best generator that minimizes the discrepancy between generated and real samples (e.g., low MMD score), this sample selection event can be characterized by

$$\text{MMD}_{inc}^2[\mathcal{F}, \boldsymbol{X}^{(k)}, \boldsymbol{Y}] \leq \text{MMD}_{inc}^2[\mathcal{F}, \boldsymbol{X}^{(m)}, \boldsymbol{Y}],$$

where $m \in \bar{\mathcal{S}}$. Then the selection event can be rewritten as

$$\boldsymbol{a}_{k,m}^\top \boldsymbol{z} \leq 0, \ \text{ for all } m \in \bar{\mathcal{S}}, \ \boldsymbol{a}_{k,m} = [0 \ \cdots 0 \underbrace{1}_{k} \ 0 \ \cdots 0 \underbrace{-1}_{m} \ 0 \cdots 0]^\top \in \mathbb{R}^S$$

and $\boldsymbol{a}_{k,m}^\top$ is a row vector of $\boldsymbol{A} \in \mathbb{R}^{(S-1) \times S}$. Under such construction, $\boldsymbol{A}\boldsymbol{z} \leq \boldsymbol{b}$ can be satisfied by setting $\boldsymbol{b} = \boldsymbol{0}$. Finally, to test the $k$-th GAN, we can set

$$\boldsymbol{\eta} = [0 \ \cdots 0 \underbrace{1}_{k} \ 0 \ \cdots 0]^\top \in \mathbb{R}^S \text{ with } \boldsymbol{\eta}^\top \boldsymbol{z} = \text{MMD}_{inc}^2[\mathcal{F}, \boldsymbol{X}^{(k)}, \boldsymbol{Y}].$$

## 3 EXPERIMENT

We trained BEGAN (Berthelot et al., 2017), DCGAN (Radford et al., 2015), STDGAN (Miyato et al., 2017), Cramer GAN (Bellemare et al., 2017), DFM (Warde-Farley & Bengio, 2016), DRA-GAN (Kodali et al., 2017), and Minibatch Discrimination GAN (Salimans et al., 2016), generated 5000 images (using Chainer GAN package [1] with CIFAR10 datasets), and extracted 512 dimensional features by pre-trained Resnet18 (He et al., 2016). For the real image sets, we subsampled 5000 images from CIFAR10 datasets and computed the 512 dimensional features using the same Resnet18. We then tested the difference between the generated images and the real images using `mmdInf` on the extracted features. We used Gaussian kernel in $\text{MMD}_{inc}$ and set the significance level to $\alpha = 0.05$.

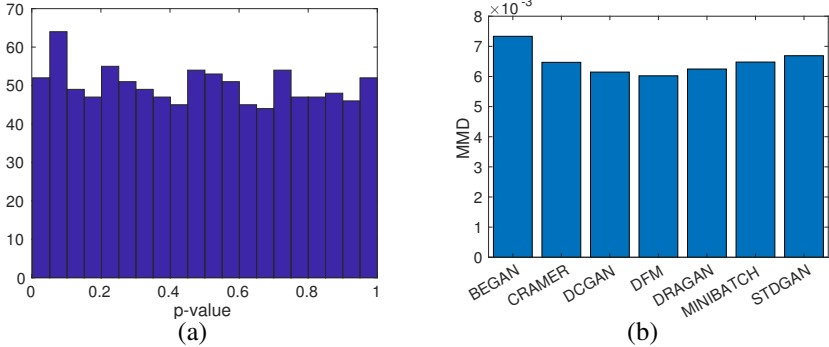

Figure 1: (a) Histogram of $p$-values over 1000 runs. (b) Averaged incomplete MMD scores.

However, we found that for all the members in the GAN family, the null hypothesis was rejected, i.e., the generated distribution and the real distribution are different. As sanity check, we evaluated `mmdInf` by constructing an "oracle" generative model that generates real images from CIFAR10. Next, we randomly selected 5000 images (a disjoint set from the oracle generative images) from CIFAR10 in each trial, and set the number of subsamples to $\ell = 5n$. Figure 1(a) showed the distribution of $p$-values computed by our algorithm. We could see that the $p$-values are distributed uniformly in the tests for the "oracle" generative model, which matched the theoretical result in Theorem 1. Thus the algorithm is able to detect the distribution difference and control the false positive rate. In other words, if the generated GANs samples do not follow the original distribution, we could safely reject the null hypothesis with a given significance level $\alpha$.

Figure 1(b) showed the estimated MMD scores of each member in GANs family. Based on the results, we could tell that DFM was the best model and DCGAN was the second best model to generate images following the real distribution. However, the difference between various members was not obvious. Developing a validation pipeline based on `mmdInf` for GANs analysis would be one interesting line of future work.

---

[1] https://github.com/pfnet-research/chainer-gan-lib

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
