# OpenReview forum: "Selecting the Best in GANs Family: a Post Selection Inference Framework"
_ICLR.cc/2018/Workshop — Accept_

### Official Review · AnonReviewer2 · 2018-03-09
**A really good paper**

**Rating:** 7
**Confidence:** 2

**Review:**

This paper tackles the problem how to evaluate the GAN model performance. The new criterion is based on U-statistics estimate of MMD. The authors described its asymptotic normality, computational efficiency and model agnosticity. The authors then propose a hypothesis testing framework based on the PSI (Post Selection Inference) and MMD for GANs analysis. The framework can find a member in GANs family with the most plausible generated samples and test whether the selected member can generate samples that cannot be differentiated from the real distribution.

Overall the paper is well-written with clear logic and accurate narratives. The methodology within the paper appears to be reasonable to me. There are also a lot of empirical results in the paper, which seems to be more than sufficient for a workshop paper. Because this is not my research area, I cannot judge its technical contribution.

One minor point: is "qualitative" in the first sentence of Introduction a typo? It seems that the authors want to say "quantitative".

---

### Official Review · AnonReviewer3 · 2018-03-09
**Evaluation/testing of GANs via maximum mean discrepancy, good fit**

**Rating:** 8
**Confidence:** 4

**Review:**

The focus of the paper is how to qualitatively evaluate the performance of GAN (generative adversarial network) techniques.
The authors propose a new MMD (maximum mean discrepancy) estimator relying on incomplete U-statistics (MMD_inc), extending the existing quadratic-time and linear-time methods. MMD_inc is asymptotically normal and efficient to compute; hence it can be adapted to hypothesis testing in the post-selection inference framework. The resulting statistical test is demonstrated on several GAN variants. The authors conclude that none of the studied 7 GAN variants can be accepted as generators (on the CIFAR10 dataset using Resnet18 feature representation).

Given the large number of recently proposed GAN methods, thanks to the principled statistical phrasing of the question and the slightly provocative nature of the the submission, it can be of wide interest to the community.  The paper is well-organized, clearly written; it can lead to lively discussions.

---

### Official Review · AnonReviewer1 · 2018-03-10
**An interesting work with practical insights**

**Rating:** 7
**Confidence:** 3

**Review:**

This work proposes a framework to evaluate the performance of GANs  based on the incomplete U-statistics estimate of maximum mean discrepancy, which measures the distribution discrepancy between generated and real examples.

The studied problem is interesting. The method may help to select the best GAN model.

---

### Decision · Program_Chairs · 2018-03-20
**ICLR 2018 Workshop Acceptance Decision**

**Decision:**

Accept

**Comment:**

Congratulations, your paper was accepted to the ICLR workshop.